# Persistence of Untreated Bed Nets in the Retail Market in Tanzania: A Cross-Sectional Survey

**DOI:** 10.3390/tropicalmed10060175

**Published:** 2025-06-19

**Authors:** Benjamin Kamala, Dana Loll, Ruth Msolla, David Dadi, Peter Gitanya, Charles Mwalimu, Frank Chacky, Stella Kajange, Mwinyi Khamis, Sarah-Blythe Ballard, Naomi Serbantez, Stephen Poyer

**Affiliations:** 1PMI Vector Control Project, Johns Hopkins University Center for Communication Programs, Dar es Salaam 14113, Tanzania; rmsolla@yahoo.com (R.M.); daviddadi1970@gmail.com (D.D.); 2School of Public Health and Social Sciences, Muhimbili University of Health and Allied Sciences, Dar es Salaam 65001, Tanzania; 3PMI Vector Control Project, Johns Hopkins University Center for Communication Programs, Baltimore, MD 65001, USA; dana.kathryn.loll@gmail.com; 4National Malaria Control Program, Ministry of Health, Dodoma 40478, Tanzania; mpgitanya2006@yahoo.com (P.G.); dismasi@yahoo.com (C.M.); frankchacky@gmail.com (F.C.); 5President’s Office, Regional Administration and Local Government, Dodoma 41185, Tanzania; kajangestella@yahoo.com; 6Zanzibar Malaria Elimination Program, Zanzibar 407, Tanzania; mwikha@hotmail.com; 7U.S. President’s Malaria Initiative, U.S. Centers for Disease Control and Prevention, Dar es Salaam 11102, Tanzania; 8U.S. President’s Malaria Initiative, U.S. Agency for International Development, Dar es Salaam 14111, Tanzania; naomiserbantez@gmail.com; 9PMI Vector Control Project, Tropical Health, Baltimore, MD 21205, USA; stephen.poyer@gmail.com

**Keywords:** insecticide-treated net, counterfeit, retail survey, leaked nets, commercial sector, mosquito net, legitimate ITN

## Abstract

The private sector in Tanzania has played an essential role in improving coverage and access to mosquito nets. This follow-up study assessed the overall market share for untreated and insecticide-treated nets (ITNs) and misleading or counterfeit ITN products in commercial markets. This study was conducted from March to April 2024 in ten regions in Tanzania. The study used mixed methods: (1) a quantitative survey among sampled outlets supported by photographic documentation of all net products and (2) key informant interviews of retailers and wholesalers. We assessed the relationship between market share and population access using ANOVA and Pearson correlation. No counterfeit or misleading nets were found, consistent with results from 2017, 2021, and 2022 surveys. Untreated nets dominated all markets, comprising 99% of all products observed and 99% of estimated net sales 3 months before the survey. Legitimate ITNs were crowded out from the studied markets. Leaked nets from free distributions were present but extremely limited (1%) and at their lowest level of the survey rounds. Untreated nets were more expensive than leaked ITNs for both regular- and queen-size nets. Despite ongoing efforts, increasing the share of legitimate ITNs remains a significant challenge in a context of large-scale public sector distributions.

## 1. Introduction

Vector control interventions have contributed to a reduction in malaria transmission over the past two decades in Tanzania, from a prevalence of 18% in 2007 to 14% in 2015–2016, and to 8% in 2022 [1]. Several studies have shown the value of insecticide-treated nets (ITNs) in preventing malaria [2]. Universal ITN distribution throughout the country—meaning distributing one ITN for every two people in a household—is one of the two core interventions that have primarily contributed to this reduction. The Government of Tanzania, through the National Malaria Strategic Plan (NMSP) and Zanzibar Malaria Elimination Program (ZAMEP), has set a target of 80% of the population having access to an ITN by 2025. Tanzania has stratified the country at the council level by the intensity of the transmission, where four main transmission strata, namely, “very low”, “low”, “moderate”, and “high”, have been identified [3,4]. However, in 2022, the Demographic and Health Survey reported that population access was 53%, far below the 80% target [1]. In Tanzania, ITNs are distributed through public sector channels, including reproductive and child health clinics for pregnant women and infants, school net programs, and mass campaigns. The most recent targeted mass campaign, conducted prior to the Tanzania Demographic and Health Survey-Malaria Indicator Survey (TDHS-MIS) 2022, took place in 52 councils on the mainland and all districts of Zanzibar in 2020. School-based distribution was performed in 14 high-priority regions, and distribution in reproductive and child health clinics in all regions was conducted annually. The most recent targeted mass campaigns in the three regions of Simiyu, Mwanza, and Geita reached 99% of households. The ITN population access will be assessed by subsequent household surveys, such as School Malaria Parasitemia Surveys and Malaria Indicator Surveys.

The private sector has played a crucial role in improving coverage and access to mosquito nets, particularly where public sector channels have not achieved the goal of one net for every two people. Additionally, the private sector has the advantage of offering additional variation in net attributes, such as shapes, sizes, and colors, not offered by public sector ITNs. The 2017–2018 Malaria Indicator Survey reported that 17% of bed nets present in households had been purchased nationwide (58% being ITNs and 42% being untreated nets) [5]. Of these, two-thirds (66%) were purchased by households in urban areas and one-third (33%) in rural areas. In the 2022 TDHS-MIS, the percentage of nets purchased increased to 23.7% nationwide (half being ITNs and half untreated nets) with a similar urban–rural distribution as in 2017 [1].

Untreated bed nets comprised 7% of all nets observed during the 2017–2018 Tanzania Malaria Indicator Survey (TMIS) [5]. In the 2022 TDHS-MIS, the percentage of untreated observed nets increased to 15%. While recognizing that untreated bed nets can fill some gaps in household net access without contributing to insecticide resistance, the National Malaria Control Program’s (NMCP) goal is to increase ITN sales while decreasing the number of untreated nets [6]. The Tanzania National Long-Lasting Insecticide-Treated Nets Strategy for Mainland (2019–2024) includes a commercial sector component, reflecting the high ownership rates of purchased nets, particularly in urban areas. Most of these purchased nets are untreated or, if treated, are likely to be “leaked” nets from public campaigns sold in the private market [7,8]. However, the private sector’s sale of ITNs has not been prioritized in Zanzibar’s National Malaria Strategic Plan.

In 2019, Innovation to Impact conducted a Selected Country Registration Process for Vector Control Tools to landscape ITN national registration processes. It was found that Tanzania required ITN manufacturers to conduct local, complete, or semi-field trials to confirm the effectiveness of ITN products, hence delaying the registration process [9]. Additionally, tax and tariff policies for mosquito netting have historically favored the importation of untreated compared with treated netting, making local production of ITNs less cost-effective. However, the registration process has improved recently, and in June 2024, 17 ITN products prequalified by WHO have been registered with the Tanzania Plant Health and Pesticides Authority (TPHPA) [10]. Tanzania has the continent’s only factory producing ITNs for both local and international markets, operating under a licensing agreement with Sumitomo Chemical. However, untreated nets continue to dominate the Tanzanian bed net market [7].

Private sector net market studies were conducted in 2017, 2021, and 2022. These studies found the sampled retail markets dominated by untreated nets, with 75%, 99%, and 88% of the market share held by such nets in 2017, 2021, and 2022, respectively [11]. Leaked ITNs from the public sector comprised 0.3% of the market share in 2021 and 8.3% in 2022. Markets in the Kigoma region had the highest frequency of leaked ITN products. Legitimate ITNs were scarce, despite the presence of a local ITN manufacturer. Three untreated nets, fabricated in China and claiming to be ITNs, were observed in 2022. The PMI Vector Control Project conducted a follow-up study in 2024 in an expanded set of retail markets to update estimates of private sector availability of mosquito nets by net type and origin, private sector net sales and market share, net prices, and factors associated with sales in different outlets and examine whether net sales varied by transmission strata. Together, results from these four surveys can inform an assessment of the commercial sector component of the Tanzania National LLIN Strategy for Mainland, which aims to increase private sector sales of ITNs to 300,000 annually by 2025.

## 2. Materials and Methods

### 2.1. Study Sites

This study was the fourth iteration of data collection in Tanzania, following survey rounds conducted using the same methodology in 2022, 2021, and 2017 [11]. It was conducted in 20 councils in ten regions, purposefully selected based on NMCP and ZAMEP recommendations about the location of major bed net markets within mainland Tanzania and Zanzibar (the same criteria used for previous net market assessments in Tanzania). The Kagera and Tanga regions were included in the study for the first time, alongside the eight previously visited regions for the 2022 study (Figure 1).

### 2.2. Study Design and Sampling

The basic design was an exploratory, observational study involving the collection of both quantitative and qualitative data, with triangulation of findings from the two components. Within each region, up to three predominantly urban local councils were purposively selected based on consultation with the local Ministry of Health (MOH) and local government and regional authority staff, taking into account the location of the largest retail market centers in each council. Within each of the twenty councils, several sub-markets were identified based on local informant knowledge of key trading areas within each council. Table 1 shows the list of 114 included markets in the 2024 study.

Within each market, a purposive sampling approach was employed to select a diverse range of net-selling outlets. Field teams initiated their surveys at the first identified net-selling outlet and then systematically navigated the market to ensure comprehensive coverage. All outlets in a given market, including mobile vendors, were visited and asked if they stocked retail nets; subsequently, an interview was conducted.

### 2.3. Study Procedures

Ten teams of three to four field workers (interviewers and supervisors) were recruited, each covering one region. Field workers were trained over 2 days following a curriculum that included applying the outlet survey questionnaire, qualitative interviewing techniques, survey administration, research ethics, and identifying various ITN and untreated net brands and suspicious and counterfeit ITN products. Staff were provided with a visual aid to help identify suspicious nets based on findings from previous survey rounds. Local authorities and relevant market associations were informed of the study and its objectives before fieldwork. Fieldwork was conducted from 17 to 29 March 2024. Research assistants collected information about outlet characteristics, net products, sales volumes, and prices using a structured questionnaire on Android tablets with Kobo Toolbox v.2.024.18 (Kobo, Cambridge, MA, USA), an Open Data Kit (ODK)-based software for mobile data collection. Data collection included taking pictures of the back and front of the packaging for each net product and capturing details of manufacturer and insecticide content (when information was present on the packaging). The mosquito net brand section also asked about the price in local currency of rectangular nets of regular, queen, and king size (small, medium, and large). These were used as the size specification varies between sites and shops. Generally, a rectangular “regular” size is equivalent to a 130 cm width net; “queen”, 160 cm; and “king”, 190 cm. Data were sent to a secure server daily and screened, with immediate feedback to the field teams as necessary.

### 2.4. Data Management and Analysis

Data were downloaded from the Kobo Toolbox secure server and imported into the statistical software package Stata v18 (StataCorp, College Station, TX, USA) for processing and analysis. First, variables were labelled using a standardized code developed for the earlier survey rounds. Data were checked for duplicate records and then cleaned, using the packaging photos to arrive at data cleaning decisions. Finally, the pictures of each net product were reviewed against other captured data as a final quality control step before analysis. The characteristics of specific products were compared within sites, and a final classification was made according to the following five category definitions based on WHO recommendations for medicines (Figure 2). Figure 2 presents the definitions of the different types of ITNs used for this study to classify the nets. The first three (counterfeit nets, nets with misleading labeling, and leaked nets) represent “problematic” net products.

During data preparation, queries arose regarding specific net products or batch numbers, prompting manufacturers to be contacted for further clarification. This enabled the definitive categorization of net products.

Statistical analysis used standard univariable and bivariable analysis. Arithmetic means were used for continuous variables to describe the central tendency, and the *t*-test was used to compare groups, provided that the data were normally distributed. Price data were presented as medians and inter-quartile ranges, and comparisons of the central tendency over time were made using the Wilcoxon rank sum test. Proportions were compared by contingency tables, and the chi-squared test was used to test for differences in proportions. To estimate the relative market share, the number of nets reported sold in the past 3 months was transformed into categorical data using ranges of sales, and data were weighted and analyzed by using the mean of each sales category as frequency weights (categories were defined based on the question wording and response options used in earlier study rounds where this approach was used rather than the recording of the reported number of sales). Market share indicators were also generated using the integer number of nets sold in the past 3 months, and the two methods produced concurrent results (Appendix A). The sales price was captured for regular-, queen-, and king-sized rectangular nets. These were used as the size specification varies between sites and shops.

The associations between (i) total nets sold and population access to any mosquito net and (ii) total nets sold by transmission strata were assessed. The proportion of people with access to any mosquito nets was obtained from the 2022 DHS survey. The DHS Spatial Data Repository publishes modeled surfaces using standardized geostatistical methods, publicly available DHS data, and a standardized set of covariates 13. The ITN access indicator was estimated using these surface layers in the second administrative unit (“district” in Tanzania). Malaria transmission strata were also obtained from the NMCP 14,15. Total net sales by council were calculated for the 2024 survey data. A Pearson correlation test was conducted for ITN access and total net sales at the council level based on their transmission strata. An ANOVA was performed separately for malaria transmission strata and total net sales within each stratum. A Shapiro–Wilk test of normality was conducted on the residuals of the ANOVA model to confirm if the residuals followed a normal distribution.

### 2.5. Key Informant Interviews

Qualitative interviews complemented quantitative retail survey data. The same field staff conducted qualitative and quantitative data collection activities. Key informants among wholesalers and retailers were identified during fieldwork, and at each market site, the target was to conduct six interviews. The inclusion criteria were the same as those for the quantitative survey, and efforts were made to obtain a mix of participating wholesalers and retailers within a given site, with a focus on wholesalers where possible. Key informant interviews (KIIs) were conducted, and snowball sampling methods were used to identify relevant wholesalers for the interviews. Privacy was considered during the interviews, and handwritten notes were taken and summarized in English. An experienced analyst undertook qualitative analysis by screening all notes and summarizing common themes and observations based on the research questions.

### 2.6. Ethical Clearance

In Tanzania, full ethical clearance was obtained from the National Institute of Medical Research (NIMR/HQ/R.8a/Vol. IX/3622) for the mainland and the Zanzibar research institution (ZAHREC/03/REC/MAR/2021/08) for Zanzibar. Participants were informed about the study’s aims and their anonymity and confidentiality related to data collected from this study. Verbal consent was obtained for both the survey and KII. A “not human subjects research” determination was obtained for the study from the Institutional Review Board (IRB) at the Johns Hopkins Bloomberg School of Public Health, Baltimore, USA (application reference number 15658, MOD4505 for the 2024 data collection).

## 3. Results

In the 2024 survey, a total of 1469 outlets were interviewed, representing 82% of the maximum planned sample of 1800. Figure 3 presents the sample composition by outlet type and region. Across the twenty study councils, the total number of interviewed outlets ranged from 20 (Missenyi, Kagera) to 105 (Nyamagana MC, Mwanza). Markets/shops were the most common category of retailer that stocked nets (64% of all outlets interviewed), followed by mobile vendors (32%). Pharmacies and supermarkets comprised less than 2% of all outlets with nets in stock.

### 3.1. Net Types

In the 1469 outlets, 2216 product observations were made, of which 2183 (98.5%) were untreated nets (Figure 4). One (<0.1%) legitimate ITN and 32 (1.4%) leaked ITNs were found in surveyed outlets; no counterfeit or misleading nets were found in the 2024 round. The percentage distribution of observed untreated and leaked ITNs was similar in 2024, 2022, and 2021.

The type of net observed in 2024 and their distribution by sales or market share in the last 3 months are presented by region in Figure 5. All leaked ITNs were found in small shops or with mobile vendors (market hawkers). Kigoma had the highest frequency of leaked ITN products. The leaked and legitimate net brands reportedly had low sales volumes. In Kigoma, more than one in ten net sales (12%) in the past 3 months were for leaked ITNs. Dar es Salaam had the highest sales volume overall, followed by Tanga, Mtwara, and Mwanza. Untreated nets dominated the markets in all sites and comprised 99% of overall sales.

Asian-fabricated, untreated nets (nearly all Chinese in origin) dominated the markets in all sites in 2024, with relative sales share ranging from 86% in Kigoma to 100% in Kagera and Mwanza (Figure 6). Tanzanian-fabricated untreated nets, including the SafiNet brand, comprised around 10% of sales in Dodoma and Tanga and less than 2% in other regions. Legitimate ITNs were only observed for sale in Arusha (n = 1 case).

### 3.2. Net Brands

Based on information from the Tanzania Plant Health and Pesticides Authority (TPHPA), which registers pesticide products in Tanzania, 17 ITN products were registered in Tanzania at the time of the survey (Table 2). Eleven of these are WHO pre-qualified products (Olyset Net, PermaNet 2.0, PermaNet 3.0, PermaNet Dual, Interceptor, Interceptor G2, MiraNet, OlysetPlus, DuraNet, DuraNet Plus, and Veeralin). Icon Life, NetProtect, LifeNet, and DawaPlus 2.0 are no longer in production by their respective manufacturers.

Interceptor G2 (n = 11), Yahe (n = 5), Olyset Plus (n = 4), DawaPlus 2.0 (n = 3), and PermaNet 2.0 (n = 3) were the most common leaked ITNs found in Tanzania in 2024. In total, 32 leaked ITN products were observed, of which 24 were in Kigoma, 4 in Songwe, 3 in Dodoma, and 1 in Arusha. The 32 products were observed in 30 outlets: one outlet in Songwe stocked the Chitetezo Net (a Tana Netting product overbranded by an NGO for distribution in Malawi) and an Interceptor G2, and one outlet in Kigoma stocked the PermaNet Dual and an Interceptor G2. Eight leaked Interceptor G2 products were found in Kigoma and three in Songwe. Of the estimated 710 units of leaked nets sold over the previous 3 months (<1% of total net sales), most occurred in Kigoma (89%), followed by Dodoma (9%). The Yahe brand accounted for 27% of the relative sales of leaked nets, compared with 21% for Interceptor G2, followed by 16% for Olyset Plus and 12% for PermaNet 2.0.

Regarding legitimate ITNs, in 2024, one conical MiraNet product manufactured by A-Z Textile Mills was observed for sale in Arusha. The product reported 30 sales in the 3 months before the survey. No legitimate ITNs were observed for sale in 2022.

#### 3.2.1. Prices

Leaked ITNs had a median price of TSH 5000 (USD 1.91) for regular (n = 23) and TSH 6500 (USD 2.48) for queen (n = 10) sizes. Untreated nets were more expensive than both these categories, with a median price of TSH 9000 (USD 3.43) for 748 regular (double) nets (IQR: TSH 8000–10,000; USD 3.05–USD 3.82) and TSH 10,000 (USD 3.82) for 813 queen-size nets (IQR: TSH 8000–12,000; USD 3.05–USD 4.58). Figure 7 shows the median prices and inter-quartile ranges for regular and queen-size nets from 2021, 2022, and 2024 surveys. Prices for queen-size nets were generally slightly higher than for regular-size nets. For both sizes, median prices for leaked ITNs were less than those of legitimate and untreated ITNs.

Median prices for untreated regular-size nets from Asian manufacturers across the ten regions from 2021 to 2024 are shown in Figure 8. In 2021, median prices ranged from TSH 8000 (USD 3.48) in Dar es Salaam, Mwanza, and Mtwara to TSH 10,500 (USD 4.57) in Zanzibar. In 2024, median prices ranged from TSH 5000 (USD 1.91) in Kigoma (n = 109) to TSH 25,000 (USD 9.54) in Kagera (n = 16). Results for several regions suggest changes in median price between 2021 and 2024 or 2022 and 2024. Between 2022 and 2024, the distribution of prices moved to higher prices in Dodoma (*p* = 0.003) and to lower prices in Songwe (*p* = 0.025). For regions included since 2021, prices declined in Kigoma (*p* < 0.0001), increased in Dar es Salaam (*p* = 0.018) and Arusha (*p* < 0.0001), and had less strong evidence or no change in Zanzibar (*p* = 0.061), Mtwara (*p* = 0.061), and Mwanza (*p* = 0.137).

Figure 9 shows the relationship between total nets sold and population ITN access (left panel) and the distribution of nets sold by transmission strata (right panel). The analysis revealed a weak positive correlation, r = 0.107, between total nets sold and population ITN access, which was not statistically significant, t(18) = 0.456 [95% CI −0.352 to 0.525], *p* = 0.654. This indicates considerable uncertainty regarding the strength and direction of the relationship. The ANOVA results of net sales in different transmission strata showed no significant differences, indicating that the number of nets sold does not vary significantly based on transmission strata classification, F (3,16) = 1.491, *p* = 0.255. A Shapiro–Wilk normality test conducted on the residuals of the ANOVA model confirmed that the residuals followed a normal distribution.

#### 3.2.2. Qualitative Results

A total of 101 informant interviews (KII) with wholesalers, retailers, and a small number of tailors (all in Zanzibar) were completed across the ten regions. Generally, sellers reported stocking various mosquito net brands in multiple shapes, sizes, and colors. Retailers noted a strong preference among customers for inexpensive, imported bed nets from China. Afya Net, Bo Xin Net, and HM Textile Nets were most cited as preferred and best-selling brands due to their affordability, soft fabrics, and wide range of sizes, shapes, and colors. Some retailers reported that customers avoid treated nets due to concerns over potential health risks and skin irritation. Customer purchasing decisions are based on the net material, the net size, and the mesh size. Bed net sales continue to be seasonal, peaking during the rainy season and at the start of the school year, when nets are bought for students to use.

While mosquito net sales are essential for business success and contribute to overall revenue, respondents reported that they are only sometimes the primary source of profit. Nets help attract customers who may purchase other goods as well. A limited number of participants, particularly in Dar es Salaam, noted that net sales were crucial to revenue and business stability.

The Kariakoo market in Dar es Salaam, the most prominent and busiest market, covering a large area and selling a wide variety of items like fresh food, spices, clothes, electronics, and household items, was commonly named as a direct or indirect source of mosquito nets. Regional wholesalers sourced nets from Kariakoo and sold them to local retailers. Additionally, it was noted that customers generally need reliable methods to identify original nets. Some customers associate higher prices with original, quality products. Retailers depend on supplier credibility and packaging details such as batch numbers and seals to ensure authenticity. Mention of Tanzania Bureau of Standards (TBS) certification was uncommon.

Respondents strongly emphasized government regulation, quality checks, and raising customer awareness to ensure only original nets are in the market. Retailers suggested stricter enforcement and monitoring by responsible authorities, including strong investigations at ports and routine monitoring of products by TBS. Feedback mechanisms between retailers and suppliers were also suggested.

Respondents agreed with the government’s plan to increase the sale of treated nets and reduce the prevalence of malaria. Retailers highlighted the importance of raising awareness, banning substandard nets, and providing subsidies to make treated nets more affordable. Some respondents in Tanga and Mwanza noted that providing treated nets in new designs and shapes could make them more attractive to consumers.

## 4. Discussion

This study was designed to assess the market share for untreated nets and ITNs and identify any leaked, misleading, or counterfeit ITN products in purposefully selected markets in ten regions of Tanzania. The study’s first major finding is that no counterfeit or misleading nets were found in Tanzania in the 2024 survey. The second finding is that untreated nets continued dominating all markets, comprising 99% of all products observed and 99% of estimated net sales 3 months before the survey. The third finding is that legitimate ITNs are crowded out from the studied markets, consistent with 2017, 2021, and 2022 results. Eight legitimate ITNs were recorded in 2017, two in 2021, none in 2022, and one in 2024.

According to NMCP, an assessment conducted in 2014 of the potentiality of the commercial market for LLIN distribution showed that approximately 1.5 million untreated nets were being sold annually in Tanzania. It further noted that there was a demand for mosquito nets from the commercial sector even 1 year after mass distribution campaigns. NMCP’s goal was to increase ITN sales by 300,000 ITNs every year up to 1.5 million at the end of 2025. These results indicate that this goal will be missed by a wide margin. ZAMEP has no private sector ITN strategy. Interviews with retailers revealed that some household populations, especially those in the higher wealth quintiles, desired additional attributes of ITNs, including a choice of shape, color, and size. Key drivers of net choice were net fabric, net size, and mesh size, with retailers noting that customers strongly preferred inexpensive imported nets from China. As it is not illegal in Tanzania to sell a net with untreated material and considering that there is effectively no private sector market for ITNs, the NMCP and ZAMEP should consider alternative engagement strategies as they reach the end date for their first goal.

A key first step is to examine the characteristics of households purchasing untreated nets from the private sector and understand the role such nets play for households: Are they helping to fill a gap in the coverage of ITNs from public channels? Is there use restricted to a specific category of household members? This demand-side data will complement the existing supply-side findings. They will help inform responses aimed at changing individual behaviors, which may have more success than the limited market-focused strategies implemented to date. Follow-up work with target consumers could identify what it would take for them to shift from buying an untreated net to sourcing an ITN from the public or private sector. Triggers will vary based on individuals’ reasons for purchasing untreated nets and will likely require a mix of demand- and supply-side approaches, including consideration of expanding public sector distribution approaches if the current channel mix is leading to groups being persistently uncovered by ITNs. Based on formative research with consumers, high-impact social and behavior change (SBC) messages could be crafted to share accurate and compelling information on the safety of ITNs and the benefits of ITN use relative to the use of untreated nets.

The fourth finding is that leaked nets from free public sector distribution were present but extremely limited and at their lowest level of the four survey rounds. This indicates that Tanzania and neighboring countries have sound ITN accountability systems, which minimize ITN leakages through robust governance and accountability strategies. These strategies should be maintained but may need strengthening to keep the ITNs reaching the targeted population, especially in countries in the south and west, as leakage was noted mainly in Kigoma in all four survey rounds from neighboring countries.

Fifth, no counterfeit or misleading nets were found in Tanzania in the 2024 survey, consistent with the 2017, 2021, and 2022 results. Over the four surveys, misleading nets were only observed in 2017 (*n* = 30), and counterfeit nets were only observed in 2022 (*n* = 1). The misleading nets from 2017 were all examples from three product brands.

### Recommendation

Based on these 2024 findings and considering those from surveys conducted in 2017, 2021, and 2022, the following recommendations can be given. First, conduct formative research among consumers to identify factors that will trigger them to shift from buying an untreated net to sourcing an ITN from the public or private sector. Second, there remains a need to strengthen the accountability of public health sector nets in neighboring countries to reduce leakage to the commercial sector. However, it is worth noting that levels of leakage within and to Tanzania appear to be very low, given the volume of legitimate public sector ITNs procured by Tanzania and its neighboring countries. Third, given the persistent domination of untreated nets in the surveys conducted since 2017, it is recommended that NMCP and ZAMEP reconvene a stakeholder workshop with the FDA, TPHPA, Tanzania Bureau of Standards, revenue authorities, and business associations to sensitize stakeholders and their clientele to the findings and determine whether having a target of increased market share for legitimate ITNs is rational given the documented results to date. Critically, any future government action should seek to maintain and build—and not undermine—the mosquito net use culture that exists in Tanzania. The demand for untreated nets shows that personal investment in malaria vector control is a choice for some consumers. This demand could help fill inevitable gaps that will be created by the current reductions in malaria funding from international donors and the expected reduction in government vector control budgets. Against a backdrop of potential decreasing access to donor-funded ITNs, the consistent use of commercially procured untreated nets could play an increasing role in malaria control through bite prevention.

This study has some limitations. First, although some fundamental results are presented by outlet type and for all regions combined, the sample of markets and outlets was not designed to represent Tanzania statistically. The sampling approach targeted the primary hubs of mosquito net commercial activity in markets across all malaria epidemiological zones; therefore, it is plausible to assume that the study accurately captured the general national situation. The study presents regional results, and comparisons are made with previous rounds regarding regional outcomes. For the eight regions visited in 2022 and the six regions in the 2021 studies, the same councils were selected for inclusion, and many of the same markets were visited. This adds validity to the comparison of survey data over time, even if any round is not statistically representative of Tanzania. Second, there may have been some misclassification of products into the net categories used for analysis, particularly for nets with unlabeled packaging. The availability of package photos for all audited products helped minimize misclassification where labelled packaging was present. Instances of nets packaged in clear bags with no labels were relatively few (*n* = 20, or 0.9% of the total sample) and are unlikely to affect the overall results. Third, while cooperation levels remained relatively high across regions among available retailers and wholesalers, the study encountered challenges engaging key KII stakeholders, including local manufacturers. This limits the perspectives considered when informing the results of the qualitative study component (notwithstanding the fact that KIIs generated retailer-reported perceptions of consumer preferences as the KIIs did not target consumers themselves). As a result, this component reflects the thoughts and opinions of retailers and wholesalers.

## 5. Conclusions

While the NMCP and ZAMEP have prioritized efforts to increase market share for legitimate ITNs and to reduce market share for untreated nets, this remains a significant challenge in a context where there are limited efforts to make legitimate ITNs available on the retail market due to the large-scale public sector distributions of ITNs through mass campaigns, school distribution, and reproductive and child health (RCH) clinics. Considering the results from the four surveys conducted since 2017, it is clear that there is no private sector market for legitimate ITNs in Tanzania. Legitimate ITNs have rarely been observed in these studies; even local manufacturers of ITNs are not finding it worthwhile to promote and sell retail ITNs, given the competition from untreated nets and leaked ITNs. At the same time, the assured availability of public sector ITNs and the widespread availability of soft, untreated nets in a range of sizes and colors appear to meet the current consumer demand for mosquito nets; however, this demand could change to favor ITNs with improved consumer understanding of the increased effectiveness of ITNs versus untreated nets.

## Figures and Tables

**Figure 1 tropicalmed-10-00175-f001:**
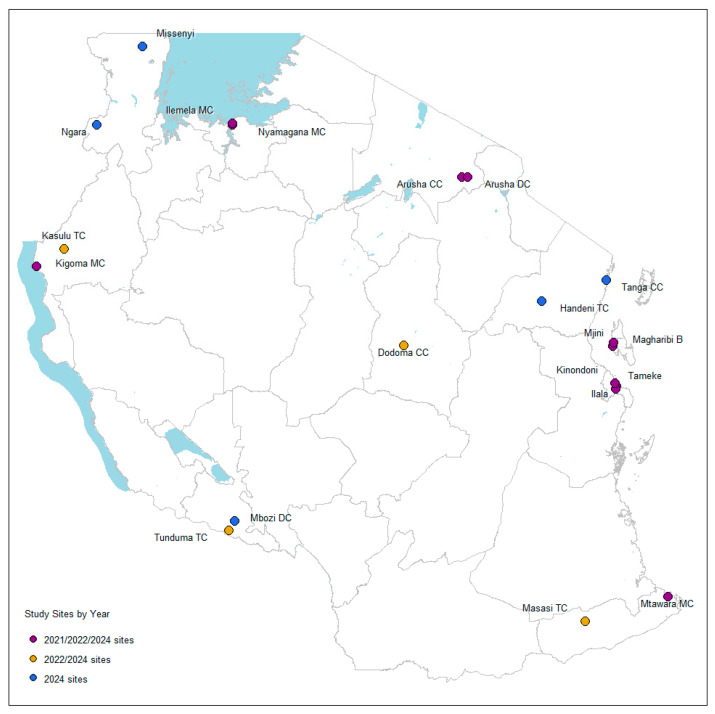
Map of study sites.

**Figure 2 tropicalmed-10-00175-f002:**
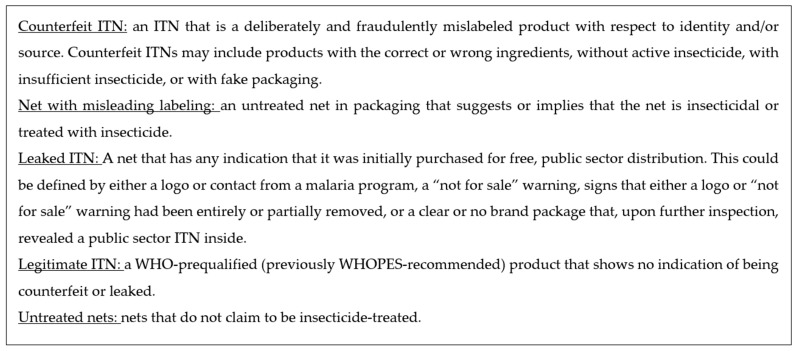
Standard definitions of net types used for the net retail study.

**Figure 3 tropicalmed-10-00175-f003:**
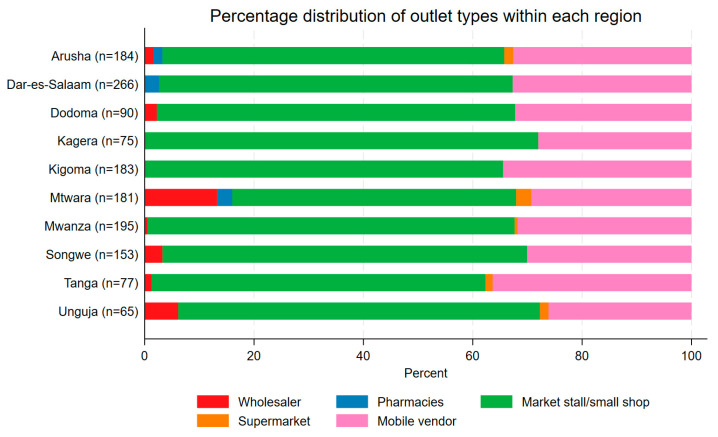
Percentage distribution of outlet types within each region in 2024 (n is the total number of outlets interviewed in the region).

**Figure 4 tropicalmed-10-00175-f004:**
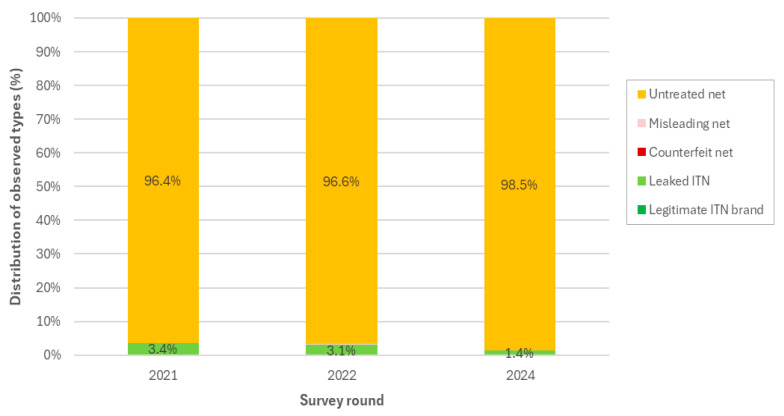
Distribution of frequency of observations by net type and survey round (2021, 2022, 2024).

**Figure 5 tropicalmed-10-00175-f005:**
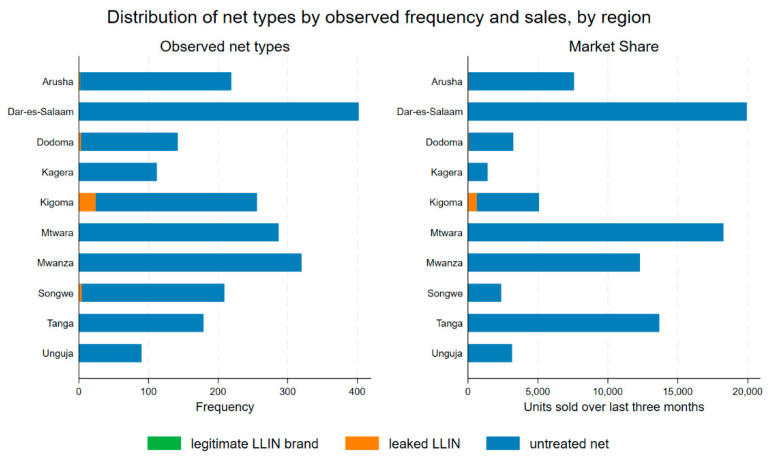
Distribution of net types by observed frequency and reported sales in 2024, by region.

**Figure 6 tropicalmed-10-00175-f006:**
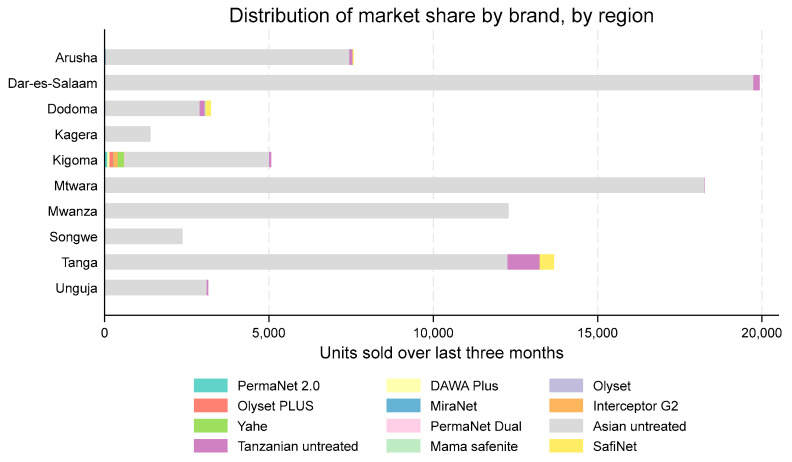
Market share by brand in 2024, by region.

**Figure 7 tropicalmed-10-00175-f007:**
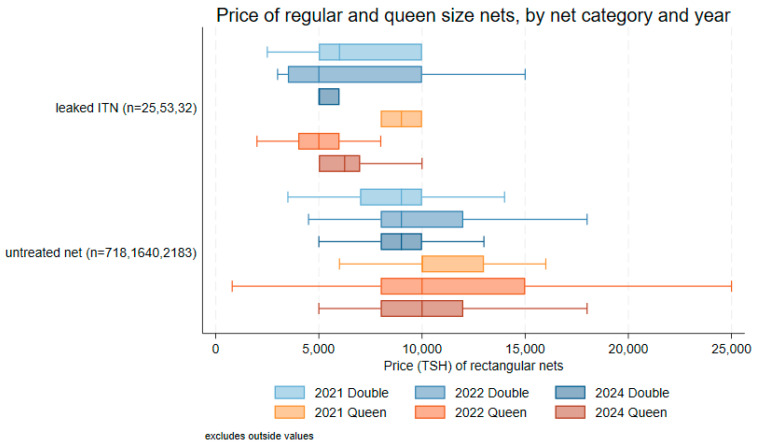
Sales prices of rectangular regular (double) and queen-size nets by net category in Tanzania in 2021, 2022, and 2024 surveys. The box plot shows the median (horizontal line), inter-quartile range (box), and adjacent values (whiskers). Outlier values are not displayed. The number of nets by category is indicated for the 2021, 2022, and 2024 surveys.

**Figure 8 tropicalmed-10-00175-f008:**
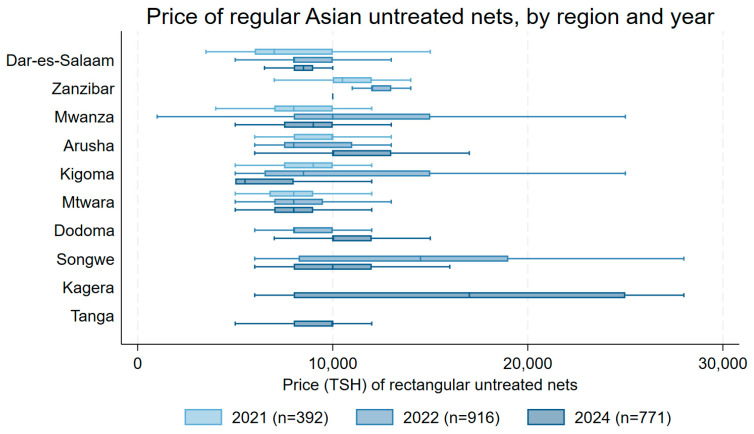
Net prices by region for regular-size Asian untreated nets.

**Figure 9 tropicalmed-10-00175-f009:**
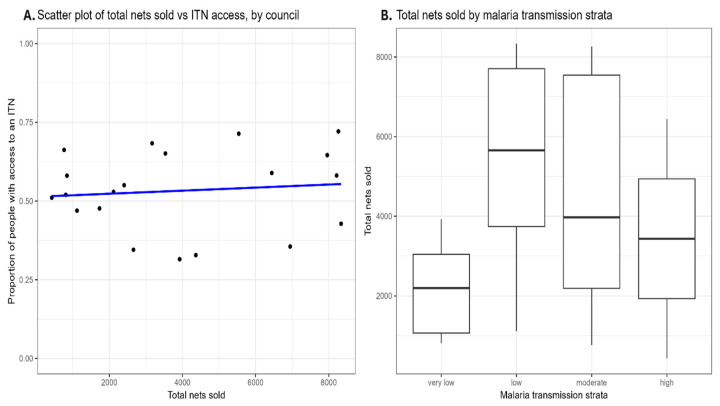
(**A**) Scatter plot of the association between population access to ITN and net sales in 20 councils. The blue line shows the linear regression result described in the text. (**B**) Distribution of council-level net sales by transmission strata.

**Table 1 tropicalmed-10-00175-t001:** Markets included at each site in the 2024 study.

N	Regions	Councils	Number of Markets	Names of Markets Visited	Outlets Visited
1	Dar-es-salaam	Kinondoni	4	Bunju, Kawe, Makumbusho, Tegeta	91
2	Dar-es-salaam	Ilala	4	Buguruni, Kariakoo, Karume, Mchikichini	86
3	Dar-es-salaam	Temeke	2	Mbagala Rangi Tatu, Tandika	89
4	Unguja	Mjini	11	Biziredi, Darajani, Kibanda Mait, Kwa Haji Tumbo, Mchangani, Mkunguni, Mlandege, Mwembeladu, Mwembeladu Magodoro, Pinda Mgongo, Soko la Sateni	33
5	Unguja	Magharibi B	11	Amani, Amani Fresh, Bububu, Bububu Kijichi, Fuoni, Fuoni/Meli 5, Magomeni, Mchina Mwanzo, Mombasa/SOS, Mombasa/Tomondo, Mwanakwerekwe	32
6	Arusha	Arusha City Council (CC)	6	Kilombero, Mjini kati, Samunge, Soko kuu, Stendi kuu, Stendi kubwa	92
7	Arusha	Arusha District Council (DC)	4	Kisongo, Ngaramtoni, Soko la usariver, Tengeru	92
8	Dodoma	Dodoma CC	9	Barabara ya 10, Barabara ya 11, Barabara ya 6, Barabara ya 9, Majengo Sokoni, Mtaa wa Tembo, Mwangaza Road, One way, Sabasaba	90
9	Mtwara	Mtwara Municipal Council (MC)	3	Mkanaledi, Nangwanda, Soko Kuu la Mtwara	90
10	Mtwara	Masasi Town Council (TC)	1	Mkuti	91
11	Songwe	Tunduma TC	3	Manzese Sokoni, Mpemba Sokoni, Soko kuu	88
12	Songwe	Mbozi DC	4	Ichenjezya Sokoni, Mlowo Sokoni, Nvozi Songwe, Vwawa Sokoni	65
13	Mwanza	Ilemela MC	12	Buswelu, Buzuruga, Igombe, Kahama center Kayenze, Kilimahewa Kiloleni, Kirumba, Kitangiri, National, Nyasaka, Pansiansi	90
14	Mwanza	Nyamagana MC	16	Buhogwa, Igogo, Karuta, Kishiri, Libeti, Lumumba, Lwagasore, Makoroboi, Market Street, Mchafu Kuoga, Mission Nata, Miti Mirefu, Mkolani, Nyerere Road, Pamba Road, Rwagasore	105
15	Kigoma	Kigoma MC	7	Bangwe, Buzebazeba, Gungu, Kigoma Mjini, Masanga, Mwanga, Seremala	96
16	Kigoma	Kasulu TC	3	Mjini, Shaya, Sofya	87
17	Kagera	Ngara	2	Kabanga Custom, Soko Kuu Ngara Mjini	55
18	Kagera	Missenyi	2	Bunazi, Kyaka	20
19	Tanga	Tanga CC	5	Barabara ya 1, Barabara ya 13, Mkwakwani, Rupia, Tangamano	37
20	Tanga	Handeni TC	5	Chogo Stand, Kivesa, Kwa Mngumi, Kwa Skanda, Soko la Zamani	40
	Total		114		1469

**Table 2 tropicalmed-10-00175-t002:** ITN products registered in Tanzania as of March 2024.

Brand	ITN Registered with WHO PQT (WHO PQT Ref. Number)	Manufacturer	Registration Number
DawaPlus 2.0 (deltamethrin and piperonyl butoxide)	No	Tana Netting, Bangkok Thailand	IN/0810
DuraNet (alphacypermethrin)	Yes (006-001)	Shobikaa Impex Private Limited, Tamil Nadu, India	IN/0889
DuraNet Plus (alphacypermethrin and piperonyl butoxide)	Yes (006-003)	Shobikaa Impex Private Limited, Tamil Nadu, India	IN/1134
Icon Life (deltamethrin)	No	Syngenta, Basel, Switzerland	IN/0333
Interceptor (alphacypermethrin)	Yes (002-001)	BASF AGRO B.V. Arnhem (NL) Freienbach Branch, Germany	IN/0381
Interceptor G2 (alphacypermethrin and chlorfenapyr)	Yes (002-002)	BASF AGRO B.V. Arnhem (NL) Freienbach Branch, Germany	Not available
LifeNet (deltamethrin)	No	Bayer Crop Science, Monheim am Rhein, Germany	IN/0641
Miracombi	No	AtoZ Group Limited, Arusha, Tanzania	Not available
MiraNet (alphacypermethrin)	Yes (009-001)	AtoZ Group Limited, Arusha, Tanzania	IN/0575
NetProtect (deltamethrin)	No	Intelligent Insect Control, France (No longer trading)	IN/0577
Netto (deltamethrin)	No	Mukpar Tanzania Ltd., Kilimanjaro, Tanzania	IN/0344
Olyset Net (permethrin)	Yes (001-004)	AtoZ Group Limited, Arusha, Tanzania	IN/0264
Olyset Plus (permethrin and piperonyl butoxide)	Yes (001-005)	AtoZ Group Limited, Arusha, Tanzania	IN/0761
PermaNet 2.0 (deltamethrin)	Yes (005-001)	Vestergaard Frandsen, Lausanne, Switzerland	IN/0278
PermaNet 3.0 (deltamethrin and piperonyl butoxide) ^1^	Yes (005-002)	Vestergaard Frandsen, Lausanne, Switzerland	Not available
PermaNet Dual (deltamethrin and chlorfenapyr)	Yes (P-03228)	Vestergaard Frandsen, Lausanne, Switzerland	Not available
Veeralin (alphacypermethrin and piperonyl butoxide)	Yes (014-002)	VKA Polymers Pvt Limited, Karur, India	Not available

^1^ During the 2022 study, Vestergaard reported that PermaNet 3.0 had been registered in Tanzania since 2020.

## Data Availability

The datasets used and/or analyzed during the current study are available from the corresponding author upon reasonable request.

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
