# Peer review of "Persistence of Untreated Bed Nets in the Retail Market in Tanzania: A Cross-Sectional Survey"

_tropicalmed, 2025, doi:10.3390/tropicalmed10060175_

Round 1

Reviewer 1 Report

Comments and Suggestions for Authors

Please find the comments in the attachment.

Reviewer 2 Report

Comments and Suggestions for Authors

Reviewer comments:

Persistence of untreated bed nets in the retail market in Tanzania: a cross-sectional survey

Thank you for the opportunity to review this manuscript. This is a simple, well-written manuscript looking at the persistence of untreated bed nets in the retail market in Tanzania, which is critical in bed net policy formulation and essential for Malaria control. The background offers the readers an opportunity to understand the problem at hand and justify the work. However, the work can be improved.

Major comments

  1. The background concentrates on Mainland Tanzania with no mention of Zanzibar. However, the surveys covered Zanzibar as well. The authors should consider highlighting this in the background as well.
  2. A little more detail on how outlets were selected for interviews in each market is important given the diverse outlet types presented in the results. This is vaguely described in lines 139-141 but should be improved. This is critical for the validity and generalization of the results.
  3. The authors need to be very clear on the statistical analysis methods used.

Minor comments

Introduction

  1. Line 45 and 46: ITN defined two times.
  2. Line 69: Is the 4 after the word “nationwide” a mistake?
  3. Line 71: TDHS-MIS abbreviations not defined before use. Check on all abbreviations

Methods

  1. Line 116: Is the 11 misplaced? Maybe these are references which are not properly incorporated or moved during formatting. Please check.
  2. Line 117: ZAMEP not pre-defined
  3. Line 200: Do you mean the medians were compared using the Wilcoxon rank-sum test/ Mann-Whitney tests (which are comparable to t-test)? Kruskal Wallis is comparable to the ANOVA test. The authors need to be very clear about the analysis method used.
  4. Lines 210-211: The description of bed sizes are repeated. This information is captured in lines 159-161.
  5. The number of KIIs in lines 233-234 can be deleted and just reported in the results (line 375).

Results

  1. Table 1: Is it possible to add the number of outlets per council? This will help strengthen Figure 3 as well.
  2. Figure 5: The market share may be driven by the population. Is it possible to use population weights for market share?
  3. Consider changing the colors by year for Figure 7 and 8 for color blind readers. I struggled in distinguishing the colors.

Discussion

  1. Line 421: I would be careful about missing the target with a wide margin. Unless I missed it, the presented results do not give the total number of nets sold in the whole country. This is partly covered in the limitations but can be thoroughly discussed.
  2. The authors use the terms study and report interchangeably. Please standardize for publication.
  3. What key policy implications do the authors suggest to the control programs given the persistence of untreated nets. This is not well, fully or clearly articulated in the recommendations section.

Round 2

Reviewer 1 Report

Comments and Suggestions for Authors

The authors have addressed all reviewers' comments and recommendations.